# The Tryptophan Index Is Associated with Risk of Ischemic Stroke: A Community-Based Nested Case–Control Study

**DOI:** 10.3390/nu16111544

**Published:** 2024-05-21

**Authors:** Dong Liu, Yan Hong, Zhenting Chen, Yifan Ma, Shangyu Xia, Shujun Gu, Hui Zuo

**Affiliations:** 1School of Public Health, Suzhou Medical College of Soochow University, 199 Ren’ai Rd., Suzhou 215123, China; dongliusph@ntu.edu.cn; 2School of Public Health, Nantong University, Nantong 226019, China; hongyan@stmail.ntu.edu.cn; 3Suzhou Medical College of Soochow University, Suzhou 215123, China; zhentingchen2022@163.com (Z.C.); myf18912606157@hotmail.com (Y.M.); xiashangyu8567345@hotmail.com (S.X.); 4Department of Chronic Disease Control and Prevention, Changshu Center for Disease Control and Prevention, Suzhou 215501, China; 18862108157@163.com; 5Jiangsu Key Laboratory of Preventive and Translational Medicine for Major Chronic Non-Communicable Diseases, Suzhou Medical College of Soochow University, Suzhou 215123, China; 6MOE Key Laboratory of Geriatric Diseases and Immunology, Suzhou Medical College of Soochow University, Suzhou 215123, China

**Keywords:** tryptophan, tryptophan index, competing amino acids, ischemic stroke

## Abstract

Background: The relative availability of the essential amino acid tryptophan in the brain, as indicated by the tryptophan index, which is the ratio of tryptophan to its competing amino acids (CAAs) in circulation, has been related to major depression. However, it remains unknown whether tryptophan availability is involved in the pathogenesis of ischemic stroke. Aims: We aimed to investigate the relationship between the tryptophan index and the risk of ischemic stroke. Methods: We performed a nested case–control study within a community-based cohort in eastern China over the period 2013 to 2018. The analysis included 321 cases of ischemic stroke and 321 controls matched by sex and date of birth. The plasma levels of tryptophan and CAAs, including tyrosine, valine, phenylalanine, leucine, and isoleucine, were measured by ultra-high-performance liquid chromatography–tandem mass spectrometry. Conditional logistic regression analyses were employed to determine incidence rate ratios (IRRs) and their 95% confidence intervals (CIs). Results: After adjustment for body mass index, current smoking status, educational attainment, physical activity, family history of stroke, hypertension, diabetes, hyperlipidemia, and estimated glomerular filtration rate, an elevated tryptophan index was significantly associated with a reduced risk of ischemic stroke in a dose–response manner (IRR, 0.76; 95% CI, 0.63–0.93, per standard deviation increment). The plasma tryptophan or CAAs were not separately associated with the risk of ischemic stroke. Conclusions: The tryptophan index was inversely associated with the risk of ischemic stroke. Our novel observations suggest that the availability of the essential amino acid tryptophan in the brain is involved in the pathogenesis of ischemic stroke.

## 1. Introduction

Stroke continues to be the leading cause of mortality in China, with the incidence rate reaching to 276.7 per 100,000 people in 2019 [1]. About 71% of total strokes worldwide are ischemic, usually manifesting as a cerebrovascular infarction [2]. The blood–brain barrier (BBB), serving as the guardian of the central nervous system, plays a crucial role in nutrient transportation, brain hemodynamics, paracellular permeability, and the development of neurological dysfunction [3].

As an essential amino acid, tryptophan must be acquired through dietary sources [4]. Notably, the availability of circulating tryptophan after penetrating the BBB determines the rate of synthesis of the neurotransmitter serotonin, also known as 5-hydroxytryptamine (5-HT), in the brain [5]. Once arrived, nearly half of the tryptophan is utilized for serotonin synthesis, whose shortage is considered to be associated with disorders of the central nervous system [6,7]. However, during its transport from circulation via the large neutral amino acid transporter system, tryptophan faces competition from other amino acids (competing amino acids, CAAs), namely tyrosine, valine, phenylalanine, leucine, and isoleucine [5,8,9]. Therefore, a higher tryptophan index (calculated as the ratio tryptophan/total CAAs) has been used as an indicator for increased tryptophan availability in the brain [5]. An animal study observed that mice being fed a diet with the lowest ratio of tryptophan to branched-chain amino acids (BCAAs: valine, leucine, and isoleucine) had central serotonin depletion and hyperphagia, which was reversed by the intervention of tryptophan or a serotonin reuptake inhibitor (SSRI) [10]. As shown in Figure 1, the initial and rate-limiting stage involves the conversion of tryptophan to the short-lived 5-hydroxytryptophan (5-HTP), followed by amino acid decarboxylation to produce 5-HT [11]. Previous studies have linked a reduced tryptophan index and decreased brain serotonin levels with depression and suicide [7,12]. A cross-sectional study suggested that tryptophan and the tryptophan index was significantly lower among acute ischemic stroke patients than their controls [13]. Notably, SSRIs are the most widely prescribed antidepressants with the ability to increase serotonin levels in synapses, compared to non-SSRIs [14]. Rasha et al. recently reported that the use of SSRIs was inversely associated with non-cardioembolic ischemic stroke risk compared with other antidepressants [15].

Despite advancements in understanding the brain’s tryptophan/serotonin completing theory published in Science, 1972 [5], there remains a knowledge gap regarding prospective association between tryptophan availability and stroke risk. Therefore, it motivated us to examine the associations of plasma tryptophan, CAAs, and the tryptophan index with the incidence of ischemic stroke using a case–control study nested within a community-based cohort.

## 2. Material and Methods

### 2.1. Study Participants

Details of the Prospective Follow-up Study on Cardiovascular Morbidity and Mortality in China (PFS-CMMC), including study strategy and enrollment, have been previously described [16,17]. In summary, after excluding individuals with severe cancer, severe disability, and/or severe psychiatric disturbance, a total of 16,457 residents aged 35 to 74 years in Changshu (Jiangsu province, China) were recruited in 2013. In the baseline survey, well-trained staff conducted face-to-face interviews with participants to collect information on demographic characteristics, education, lifestyle factors, and history of diseases using standard questionnaires. Written informed consent was obtained from each participant. Ethical approval for record linkage between the baseline survey data and study outcomes was acquired from the Ethics Committee of Soochow University (No.: SUDA20210127H01, Suzhou, China).

### 2.2. Follow-Up and Definition of Ischemic Stroke

The follow-up began with the baseline survey conducted in 2013, and continued until the date of death, occurrence of stroke, or 31 December 2018, whichever occurred first. Hospital records of discharge diagnoses and the Cause of Death Registry were used to accurately obtain the data linkage of disease or cause of death. The International Classification of Diseases, Tenth Revision (ICD-10 codes: I63 except for I63.9) was applied to define ischemic stroke [17]. If a participant experienced more than one stroke event during the follow-up period, only the first event was taken into consideration.

### 2.3. Selection of Cases and Controls

As shown in Figure 2, a total of 137 participants were excluded due to stroke history prior to enrolment, as well as 207 participants with missing baseline information. Over a median follow-up of 5.3 years, a total of 321 incident cases of ischemic stroke were identified. For each case, one control was matched by date of birth (±1 year) and sex using the incidence density sampling method [18].

### 2.4. Measurements of Biomarkers

Venous blood samples were collected from participants after a minimum of 8 h fasting overnight [16,17]. Procedures of plasma separation and preservation have been mentioned in a previous study [19]. Plasma tryptophan, tyrosine, valine, phenylalanine, leucine, isoleucine, and cotinine were detected by ultra-high-performance liquid chromatography–tandem mass spectrometry (UHPLC–MS/MS) [20,21]. All laboratory staff were blinded to the plasma case/control status. The within-day and between-day coefficients of variation were 0.70–3.50% (tryptophan: 1.85%, tyrosine: 3.50%, valine: 0.70%, phenylalanine: 1.70%, leucine: 1.40%, isoleucine: 1.25%, and cotinine: 2.47%) and 2.45–17.3% (tryptophan: 4.81%, tyrosine: 4.65%, valine: 3.70%, phenylalanine: 4.88%, leucine: 2.68%, isoleucine: 17.3%, and cotinine: 2.45%), respectively. Serum glucose levels were determined using the oxidase enzymatic method; creatinine was measured via the picric acid method; and total cholesterol (TC), triglycerides (TG), and high-density lipoprotein cholesterol (HDL-C) were assessed using enzymatic methods [22,23].

### 2.5. Covariates

We selected the covariates mainly following the basic criteria of confoundment and based on the current literature [24,25]. Body mass index (BMI) was calculated as body weight divided by the square of height, kg/m^2^. Smoking status was determined by self-report and plasma cotinine levels. Briefly, individuals with plasma cotinine levels of 85 nmol/L or above were classified as current smokers, regardless of their self-reported status [16]. Physical activity was assessed using a standardized questionnaire and estimated as metabolic equivalent-hours/day (MET-h/d) [26], including occupation, transportation, home activity exercise, etc. [27]. Educational attainment was classified into three groups: 0 years (no formal education), 1–5 years (primary school), and ≥6 years (middle school or higher). Hypertension was identified by self-report, readings of mean systolic (diastolic) blood pressure ≥140 (90) mm Hg, or the use of antihypertensive drugs. Diabetes was determined by self-report, the use of insulin/glucose-lowering drugs, or a fasting blood glucose level of ≥7.0 mmol/L. Hyperlipidemia was determined by self-report, the use of lipid-lowering treatment, or laboratory measurement (TC > 6.20 mmol/L, TG > 2.30 mmol/L, or HDL-C < 1.00 mmol/L) [28]. In addition, the equation from Chronic Kidney Disease Epidemiology Collaboration (CKD-EPI) was used to compute the estimated glomerular filtration rate (eGFR, mL/min/1.73 m^2^) [29].

### 2.6. Statistical Analyses

Continuous variables are presented as medians (interquartile ranges), while categorical variables are presented as numbers (percentages). We assessed differences between cases and their matched controls using the Wilcoxon test for continuous variables and the Chi-squared test for categorical variables. To evaluate *p* trends across quartiles of biomarkers or the tryptophan index at baseline, we used the Jonckheere–Terpstra test for all continuous variables due to their skewed distributions. For binary categorical variables, we employed the Cochran–Armitage Trend test, and the Cochran–Mantel–Haenszel test for other categorical variables.

Conditional logistic regression models were applied to assess the associations of tryptophan, its CAAs, and the tryptophan index with the risk of ischemic stroke, separately. In a nested case–control study using incidence density sampling, this method estimates incidence rate ratios (IRRs) [30]. IRRs and corresponding 95% confidence intervals (CIs) were calculated across quartiles according to the distribution of controls or per standard deviation (SD) increment of natural-log-transformed biomarkers. We constructed three models. Model 1 was an unadjusted model conditioning on the individual case set. Model 2 was adjusted for educational attainment (0 years, 1–5 years, or ≥6 years), current smoking status (yes or no), BMI (continuous), and physical activity (quartiles). Model 3 was adjusted for family history of stroke (yes or no), hypertension (yes or no), diabetes (yes or no), hyperlipidemia (yes or no), and eGFR (continuous). Moreover, we additionally included total CAAs in the multivariable model for tryptophan to check for a potential confounding effect.

Moreover, we conducted stratified analyses by sex (male or female), age (median), BMI (median), physical activity (median), hypertension (yes or no), hyperlipidemia (yes or no), and eGFR (<90 or ≥90 mL/min/1.73 m^2^) using unconditional logistic regression [16,31,32,33]. Statistical interaction was evaluated at the multiplicative scale [34]. Meanwhile, those cases that developed stroke and their matching controls in the first year of follow-up were excluded to check for potential reverse causation bias and the robustness of the primary findings. Furthermore, we calculated the E-value to examine the potential impact of unmeasured confounders [35]. A two-sided *p*-value less than 0.05 was considered to indicate statistical significance. All analyses of the data were conducted using R software, version 4.1.2 (available at https://www.r-project.org, accessed on 1 November 2021).

## 3. Results

### 3.1. Baseline Characteristics

The characteristics at baseline of 321 ischemic stroke cases and their controls are illustrated in Table 1. There was no statistical difference between the two groups in BMI, current smoking status, physical activity, or educational attainment (*p* > 0.1). The cases were more likely to have diabetes (*p* = 0.004) and hypertension (*p* < 0.001), but not hyperlipidemia. Furthermore, these participants with incident ischemic stroke had lower baseline levels of the tryptophan index (*p* = 0.010) and a higher fasting glucose (*p* = 0.013) and isoleucine (*p* = 0.038) than the controls, whereas no significant difference was observed in other amino acids, eGFR, lipid profiles, or total CAAs (all *p* > 0.05).

As presented in Appendix A, participants possessing a higher tryptophan index were generally younger (*p* trend = 0.010) and more likely to be female (*p* trend = 0.032). Moreover, no significant difference was observed in current smoking status, physical activity levels, educational attainment, or prevalence of hypertension across quartiles (all *p* trend > 0.05). Notably, individuals with higher quartiles of the tryptophan index showed a lower prevalence of diabetes (*p* trend < 0.001) and hyperlipidemia (*p* trend = 0.003), a decreased trend in TG (*p* trend = 0.017) and fasting glucose (*p* trend < 0.001), and an elevated trend in HDL-C and eGFR (*p* trend < 0.001).

### 3.2. The Tryptophan Index, Its Components, and the Risk of Ischemic Stroke

As illustrated in Table 2, the tryptophan index was found to be inversely associated with the risk of ischemic stroke yielding an adjusted IRR of 0.76 (95% CI: 0.63–0.93, *p* = 0.006) per one SD increment of log-transformed level and E-value of 1.96 (1.36). Likewise, compared with the first quartile (Q1), the adjusted IRR for the highest quartile (Q4) of the tryptophan index was 0.53 (95% CI: 0.31–0.88, *p* trend = 0.008) and the corresponding E-value was 3.18 (1.53). In contrast, there was no significant association of tryptophan with ischemic stroke risk, either as a continuous variable or by quartile. The same was true for total CAAs and the individual amino acids tyrosine, valine, phenylalanine, isoleucine, and leucine after multivariable adjustment (Appendix A). However, the association between tryptophan and the risk for ischemic stroke became significant after further adjustment for total CAAs. Compared with the Q1, the IRR was 1.02 (95% CI: 0.61–1.71) for the second quartile (Q2) [E-value: 1.16 (1.00)], 0.51 (95% CI: 0.29–0.88) for the third quartile (Q3) [E-value: 3.33 (1.53)], and 0.57 (95% CI: 0.30–1.07) for Q4 [E-value: 2.90 (1.00)]. The corresponding IRR was 0.75 (95% CI: 0.59–0.94, *p* = 0.015) per one SD increment of log-transformed level, and the E-value was 2.00 (1.32).

### 3.3. Sensitivity Analyses 

After exclusion of the cases that developed ischemic stroke within the first year, the significant association of the tryptophan index with ischemic stroke risk was still essentially unchanged (adjusted IRR: 0.51 (95% CI: 0.29–0.88), *p* trend = 0.012, Q4 vs. Q1; 0.71 (95% CI: 0.55–0.91), *p* = 0.007, per one SD increment).

### 3.4. Stratified Analyses

The findings remained consistent across various stratifications, including sex (male or female), age (median), BMI (median), physical activity (median), hypertension (yes or no), hyperlipidemia (yes or no), and eGFR (<90 or ≥90 mL/min/1.73 m^2^). No significant effect modification was observed among these stratified factors for the association between the tryptophan index and the risk of ischemic stroke (Figure 3).

## 4. Discussion

### 4.1. Principal Findings

In the present nested case–control study, we found that an elevated tryptophan index, indicating a higher availability of tryptophan in the brain, was significantly associated with a lower risk of ischemic stroke. The individual CAAs were not associated with the risk after multivariable adjustment. 

### 4.2. The Tryptophan Index and Ischemic Stroke Risk

Amino acid neurotransmitters, including excitatory amino acids (such as glutamate and aspartate) and non-excitatory amino acids (such as glycine, serine, and threonine), have been suggested to contribute the progression of ischemia [17,36]. Certainly, the homeostasis of extracellular amino acids in the central nervous system is essential for maintaining brain function. As illustrated by our recent study, the elevated ratio of glycine to lysine was associated with the decreased risk of ischemic stroke [17]. However, there is still a lack of prospective studies to systematically investigate the association between the neurotransmitter-related amino acids and stroke risk, especially for tryptophan and its CAAs. The established evidence has shown that the decreased utilization of tryptophan due to elevated circulating levels of CAAs may decrease serotonin concentration in the brain [37]. In the present study, upon additional adjustment for total CAAs, the initially observed association between tryptophan and ischemic risk became significant. Therefore, the protective effect of tryptophan on ischemic stroke might largely be offset by CAAs across the BBB. Also, our findings suggested that the elevated tryptophan index, which means a relatively high availability of tryptophan in the brain, was mainly driven by higher plasma levels of tryptophan when it was linked with the risk of ischemic stroke. These findings support dietary recommendations to increase the intake of tryptophan-rich foods, which could enhance brain health and reduce stroke risk. These insights encourage further interdisciplinary research to explore nutritional strategies for stroke prevention and treatment.

### 4.3. Mechanisms

It has been reported that the tryptophan index was a marker of immune-inflammatory response [38] and positively related to the level of anti-inflammatory interleukin-10, which may play a neuroprotective role and inhibit the reduced synthesis of 5-hydroxytryptamine in the brain [13]. Given the close relationship with serotonin, the decreased tryptophan index blunted the neurotransmitter synthesis that was associated with depression [39,40]. In addition, serotonin as a strong vasoconstrictor could innervate cerebral arteries, arterioles, and veins [41,42], whereas a weak vasoconstrictive effect due to a decreased tryptophan index might increase blood flow [42]. Moreover, the higher brain serotonin levels could attenuate platelet activation and improve the thrombotic pathway, which could decrease the risk of myocardial infarction [43], and may also relate to a reduced risk of stroke [44].

### 4.4. Strengths and Limitations

The main strengths include a comprehensive measurement of tryptophan and its CAAs, a complete follow-up, and high generalizability by recruiting community residents, which could minimize selection bias. However, the present study also has several limitations. Firstly, the residual confoundment by unknown or untested covariates cannot be excluded. For instance, we lacked data on the use of antidepressants or prior diagnoses of depression before blood sampling. Both factors are known to influence the tryptophan index [7,12,45]. However, we have excluded those residents with severe psychiatric disturbance, which may help to reduce this potential confounding effect. In addition, statin treatment may also have affected our observed associations, because of its potential regulating roles in both tryptophan metabolism and cholesterol reduction [46]. Secondly, our study did not assess the relationship between the tryptophan index and specific subtypes of ischemic stroke due to the absence of relevant data. Thirdly, we did not measure tryptophan metabolites involved in the serotonin pathway, such as serotonin itself, which restricts further investigative possibilities.

## 5. Conclusions

In the present study, we observed that an elevated tryptophan index, mainly driven by increased plasma tryptophan, was independently associated with a lower risk of ischemic stroke in community residents. In contrast, the CAAs tyrosine, valine, phenylalanine, leucine, and isoleucine were not associated. Our novel findings underscore the critical role of the availability of the essential amino acid tryptophan in the brain in the pathogenesis of ischemic stroke.

## Figures and Tables

**Figure 1 nutrients-16-01544-f001:**
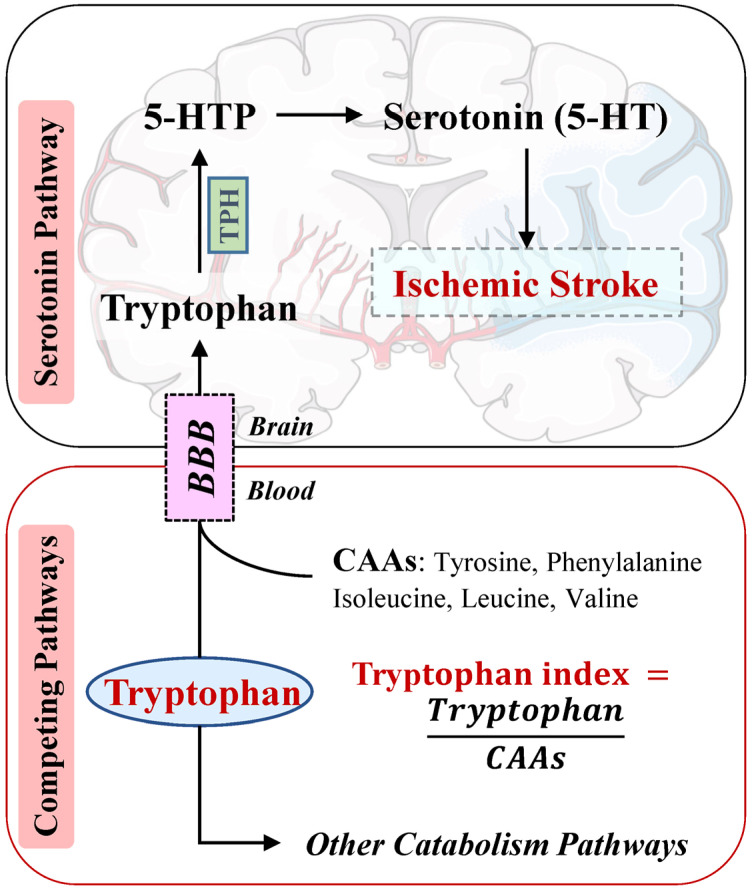
Tryptophan’s competing pathways from blood to brain. Abbreviations: 5-HT, 5-hydroxytryptamine; 5-HTP, 5-hydroxytryptophan; BBB, blood–brain barrier; CAAs: competing amino acids; TPH, tryptophan hydroxylase.

**Figure 2 nutrients-16-01544-f002:**
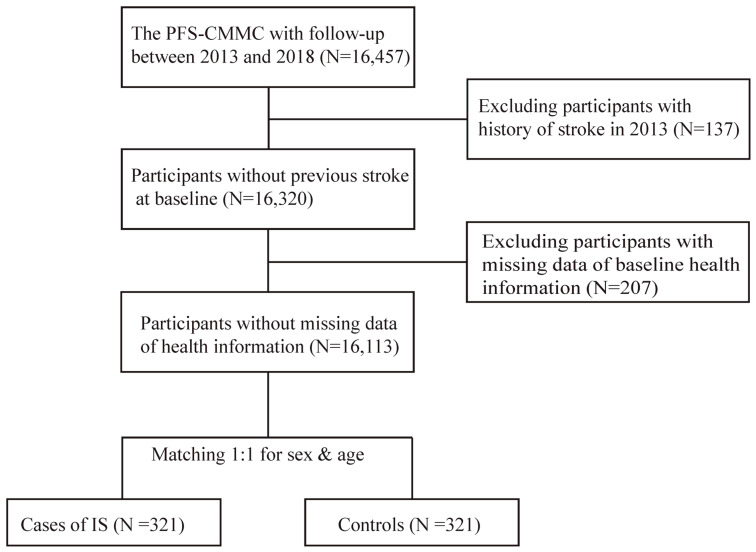
Flowchart for the nested case–control study conducted within the Prospective Follow-up Study on Cardiovascular Morbidity and Mortality in China (PFS-CMMC). Abbreviations: PFS-CMMC, the Prospective Follow-up Study on Cardiovascular Morbidity and Mortality in China; IS, ischemic stroke.

**Figure 3 nutrients-16-01544-f003:**
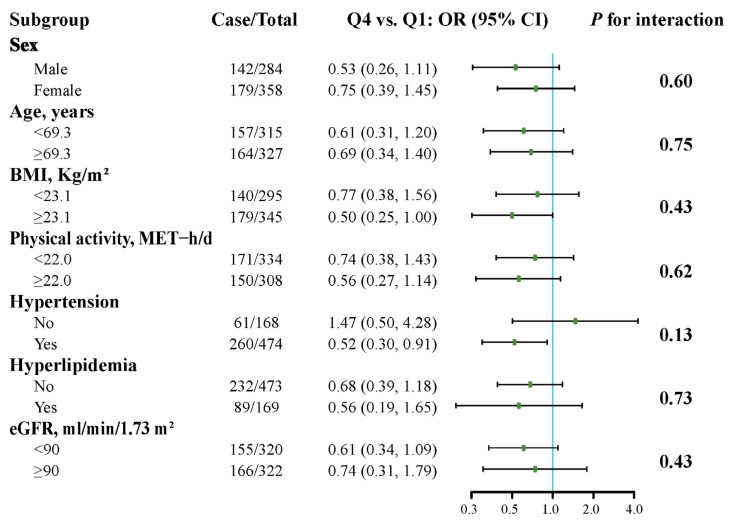
Stratified analyses for the association between tryptophan index and risk of ischemic stroke. The model was adjusted for age (continuous), sex (male or female), BMI (continuous), current smoking status (yes or no), educational attainment (0 year, 1–5 years, or ≥6 years), physical activity (quartiles), family history of stroke (yes or no), hypertension (yes or no), diabetes (yes or no), hyperlipidemia (yes or no), and eGFR (continuous). Abbreviations: BMI, body mass index; eGFR, estimated glomerular filtration rate; MET, metabolic equivalent.

**Table 1 nutrients-16-01544-t001:** Demographic and clinical characteristics at baseline between ischemic stroke cases and controls (*n* = 642).

	Cases (*n* = 321)	Controls (*n* = 321)	*p*-Value
Age (years)	69.5 (63.2, 75.2)	69.6 (63.4, 75.1)	
Male (%)	142 (44.2)	142 (44.2)	
BMI (kg/m^2^)	23.6 (21.4, 25.9)	23.3 (21.4, 25.7)	0.53
Currently smoking (%)	83 (25.9)	80 (24.9)	0.79
Physical activity (MET-h/d)	20.4 (12.3, 34.8)	21.8 (13.5, 34.4)	0.37
Educational attainment (%)			0.85
0 year	157 (49.1)	153 (47.7)	
1–5 years	121 (37.8)	121 (37.7)	
≥6 years	42 (13.1)	47 (14.6)	
TC (mmol/L)	4.95 (4.30, 5.61)	4.84 (4.21, 5.51)	0.24
TG (mmol/L)	1.32 (0.95, 1.88)	1.23 (0.92, 1.70)	0.23
HDL-C (mmol/L)	1.40 (1.16, 1.69)	1.48 (1.21, 1.71)	0.05
Fasting glucose (mmol/L)	5.54 (5.07, 6.23)	5.40 (4.98, 5.96)	0.013
Diabetes (%)	61 (19.0)	35 (10.9)	0.004
Hypertension (%)	260 (81.0)	214 (66.7)	<0.001
eGFR (mL/min/1.73 m^2^)	84.5 (72.4, 91.0)	85.4 (74.3, 93.2)	0.06
Hyperlipidemia (%)	89 (27.7)	80 (24.9)	0.42
Tryptophan (μmol/L)	76.6 (66.2, 87.9)	79.2 (67.0, 90.3)	0.19
Tyrosine (μmol/L)	89.5 (77.3, 106)	91.4 (80.0, 107)	0.23
Valine (μmol/L)	268 (237, 301)	264 (233, 296)	0.20
Phenylalanine (μmol/L)	69.3 (61.4, 80.1)	69.6 (62.2, 79.9)	0.83
Isoleucine (μmol/L)	86.8 (73.3, 106)	83.7 (71.1, 99.4)	0.038
Leucine (μmol/L)	115 (100, 134)	114 (100, 129)	0.29
Total CAAs (μmol/L)	636 (569, 700)	626 (561, 696)	0.28
Tryptophan index (×100)	12.1 (10.7, 13.6)	12.5 (11.3, 13.8)	0.010

The differences between cases and controls were assessed using Chi-squared tests for categorical variables, presented as a number and percentage, and Wilcoxon test for continuous variables, presented as a median and interquartile range. Abbreviations: BMI, body mass index; eGFR, estimated glomerular filtration rate; HDL-C, high-density lipoprotein cholesterol; MET, metabolic equivalent; TC, total cholesterol; TG, triglyceride.

**Table 2 nutrients-16-01544-t002:** Incidence rate ratio (IRR) and 95% confidence intervals (CIs) for ischemic risk by conditional logistic regression models.

	Cases/Controls	Model 1	Model 2	Model 3
		IRR (95% CI)	IRR (95% CI)	IRR (95% CI)
Tryptophan index (×100)				
Q1 (<11.2)	101/81	1.00 (Ref)	1.00 (Ref)	1.00 (Ref)
Q2 (11.3, 12.5)	88/81	0.87 (0.56, 1.34)	0.86 (0.55, 1.33)	0.91 (0.58, 1.45)
Q3 (12.6, 13.7)	68/78	0.68 (0.43, 1.07)	0.62 (0.39, 1.01)	0.63 (0.38, 1.05)
Q4 (>13.8)	64/81	0.62 (0.40, 0.98)	0.56 (0.35, 0.90)	0.53 (0.31, 0.88)
*p* for trend		0.023	0.008	0.008
Continuous	321/321	0.79 (0.67, 0.93)	0.76 (0.64, 0.90)	0.76 (0.63, 0.93)
*p*-value		0.005	0.002	0.006
Tryptophan (μmol/L)				
Q1 (<67.0)	89/81	1.00 (Ref)	1.00 (Ref)	1.00 (Ref)
Q2 (67.1, 79.1)	99/80	1.16 (0.75, 1.79)	1.16 (0.74, 1.82)	1.22 (0.76, 1.79)
Q3 (79.2, 90.3)	60/80	0.70 (0.45, 1.09)	0.67 (0.43, 1.05)	0.63 (0.39, 1.03)
Q4 (>90.4)	73/80	0.82 (0.52, 1.29)	0.75 (0.47, 1.22)	0.72 (0.44, 1.21)
*p* for trend		0.13	0.06	0.05
Continuous	321/321	0.89 (0.76, 1.05)	0.85 (0.72, 1.10)	0.83 (0.69, 1.00)
*p*-value		0.17	0.07	0.05

Model 1: Unadjusted model, based on individual case set; Model 2: adjusted for BMI (continuous), current smoking status (yes or no), educational attainment (0 year, 1–5 years, or ≥6 years), and physical activity (by quartiles), based on individual case set; Model 3: further adjusted for family history of stroke (yes or no), hypertension (yes or no), diabetes (yes or no), hyperlipidemia (yes or no), and eGFR (continuous). To normalize the distribution of the continuous tryptophan index, a natural logarithmic (log) transformation was applied. Abbreviations: BMI, body mass index; eGFR, estimated glomerular filtration rate.

## Data Availability

The data, codebook, and analytical code described in this manuscript will be accessible upon request, subject to approval by the corresponding author.

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
