# Peer review of "The Tryptophan Index Is Associated with Risk of Ischemic Stroke: A Community-Based Nested Case–Control Study"

_nutrients, 2024, doi:10.3390/nu16111544_

Round 1

Reviewer 1 Report

Comments and Suggestions for Authors

Dear Authors

I had the pleasure of reading the manuscript "The tryptophan index is associated with risk of ischemic stroke: a community-based nested case-control study". The article is interesting but there are some issues that still deserve further attention. 

1. Do the authors know if all cases of stroke were confirmed with neuroimaging?

2. Did the authors estimate the sample size before the enrollment? 

3. When exactly were the biomarkers measured? Was it standardized? 

4. Did the authors include any patient with a previous history of brain disease other than stroke, e.g. depression? 

5. Did the authors consider to perform a survival analyses? 

6. The authors did not assess the characteristics of an eventual statin therapy. This must be recognized as a limitation with an appropriate reference. 

Author Response

Reviewer 1: Comments and Suggestions for Authors

Dear Authors

I had the pleasure of reading the manuscript "The tryptophan index is associated with risk of ischemic stroke: a community-based nested case-control study". The article is interesting but there are some issues that still deserve further attention. 

  1. Do the authors know if all cases of stroke were confirmed with neuroimaging?

Response: Yes, all of the stroke cases were confirmed using neuroimaging, specifically computed tomography (CT) or magnetic resonance imaging (MRI) brain imaging scans, as per the diagnostic criteria outlined by the World Health Organization (WHO). Clinical symptoms and signs of stroke were also considered in conjunction with the neuroimaging results to ensure an accurate diagnosis.

  1. Did the authors estimate the sample size before the enrollment? 

Response: We did not calculate accurately the sample size of this nested case control study before the enrollment, as it was based on a large-scale cohort study. However, the power calculation can be conducted post-hoc to assess the adequacy of the sample size for detecting significant effects.

Given the parameters listed as follows: Odds ratio =0.53 (Q4 versus Q1, Table 2 for ischemic stroke), percentage exposed among controls =25% (the Q4 of continuous Tryptophan index defined as exposure), α=5%, number of pairs =312 (http://sampsize.sourceforge.net/iface/s3.html), we calculated the study power of 88.3%, which can basically meet the design requirements.

  1. When exactly were the biomarkers measured? Was it standardized?

Response: The EDTA plasma samples collected at the baseline survey in 2013 were stored at -80℃ all the time until the assay. The biomarkers were measured by UHPLC-MS/MS in Dec 2020. According to a recent report, compared to shorter storage samples (<4 years), only ~2% of total metabolites and 5% of amino acids in plasma stored between 4-7 years can be altered [1]. Furthermore, we deemed standardized procedures and quality control as of vital importance in the measurement of the metabolites. The laboratory staff was blinded to the case-control status and other clinical data of the blood samples. One quality control sample was mixed per ten test samples in the UHPLC-MS/MS platform.  The within-day and between-day coefficients of variation indicated the measurements of the biomarkers were reliable.

  1. Did the authors include any patient with a previous history of brain disease other than stroke, e.g. depression?

Response: This cohort at baseline excluded individuals with severe cancer, severe disability, or severe psychiatric disorders. However, we had no information about depression status or antidepressants use. This limitation has been mentioned in our paper, please see Line 275-279.

“However, the present study has also several limitations. First, the residual confounding by unknown or untested covariates cannot be excluded. For example, we had no information on antidepressants use or depression diagnosis before blood sampling, both of which have been linked to the tryptophan index (7, 12, 45).”

  1. Did the authors consider to perform a survival analyses? 

Response: Thank you for your suggestion. We appreciate your insightful feedback. Survival analysis is indeed a valuable statistical method for assessing time-to-event outcomes. However, the nested case-control study is an efficient and economic approach for assessing exposure-disease associations particularly in large cohorts. In the present study, we applied the incidence density sampling to match controls for each case on a timescale that could provide unbiased results [2]. We will carefully consider the feasibility and appropriateness of conducting survival analysis in our future studies.

  1. The authors did not assess the characteristics of an eventual statin therapy. This must be recognized as a limitation with an appropriate reference.

Response: Thank you for this helpful suggestion. We acknowledge that the characteristics of statin therapy were not assessed in our study, which is an important limitation. We have included a discussion of this limitation in the revised manuscript. Please see Line 283-285.

“In addition, statin treatment may also have affected our observed associations, because of its potential regulating roles both in tryptophan metabolism and cholesterol reduction (46).

References:

  1. Wagner-Golbs, A.; Neuber, S.; Kamlage, B.; Christiansen, N.; Bethan, B.; Rennefahrt, U.; Schatz, P.; Lind, L. Effects of Long-Term Storage at −80 °C on the Human Plasma Metabolome. Metabolites 2019, 9, doi:10.3390/metabo9050099.
  2. Richardson, D.B. An incidence density sampling program for nested case-control analyses. Occup Environ Med 2004, 61, e59, doi:10.1136/oem.2004.014472.

Reviewer 2 Report

Comments and Suggestions for Authors

The study carried out by the authors is extremely interesting as it focuses on examining the association of tryptophan index with the risk of ischemic stroke through a case-control study nested within a community-based cohort in Changshu, China. Eastern, during the period 2013-2018. The inclusion of 321 incident cases of ischemic stroke and 321 controls matched for gender and date of birth provides a solid basis for the analysis.

The conclusions of the study are significant, as they indicate that a high tryptophan index is significantly associated with a decreased risk of ischemic stroke in a dose-dependent manner. However, no association was found between plasma tryptophan or competing amino acids separately and the risk of ischemic stroke.

The study is supported by detailed analysis and an up-to-date bibliography, which increases its credibility. However, it is noted that the discussion is quite concise and could be improved. It would be beneficial for the reader to delve deeper into the implications that this study may have, as well as the nutritional and epidemiological recommendations that could arise from these findings. Furthermore, it is suggested to address the mechanism underlying this effect in greater depth, which would enrich the reader's understanding of the biological processes involved.

One area of improvement identified is the lack of discussion on the role of non-excitatory amino acids in the pathogenesis of stroke, as well as related research in this field.

In summary, the article presents valuable and significant research in the field of the association of tryptophan with the risk of ischemic stroke. However, an expansion in the discussion is suggested to address implications, recommendations, and underlying mechanisms, as well as the inclusion of related research to contextualize the findings within the current scientific landscape.

Author Response

Response: We sincerely appreciate the comments and suggestions provided by the reviewer.

We thank the suggestion to explore the role of non-excitatory amino acids in the pathogenesis of stroke and to incorporate related research in the field to provide a more contextualized understanding of our findings within the current scientific landscape. Following the suggestion, we have now included this point in the Discussion (please see Line 244-251, 259-262).

“Amino acid neurotransmitters, including excitatory amino acids (such as glutamate and aspartate) and non-excitatory amino acids (such as glycine, serine and threonine) have been suggested to contribute the progression of ischemia (17, 36). Certainly, the homeostasis of extracellular amino acids in central nervous system is essential for keeping the brain function. As illustrated by our recent study, the elevated ratio of glycine to lysine was associated with the decreased risk of ischemic stroke (17). However, there were still lack of prospective studies to systematically investigate the association between the neurotransmitters-related amino acids and stroke risk, especially for the tryptophan and its CAAs.”

“The findings support dietary recommendations to increase intake of tryptophan-rich foods, which could enhance brain health and reduce stroke risk. These insights encourage further interdisciplinary research to explore nutritional strategies for stroke prevention and treatment.”